# Evaluation of the Synovial Effects of Biological and Targeted Synthetic DMARDs in Patients with Psoriatic Arthritis: A Systematic Literature Review and Meta-Analysis

**DOI:** 10.3390/ijms24055006

**Published:** 2023-03-05

**Authors:** Maria Sofia Ciliento, Veronica Venturelli, Natale Schettini, Riccardo Bertola, Carlo Garaffoni, Giovanni Lanza, Roberta Gafà, Alessandro Borghi, Monica Corazza, Alen Zabotti, Sonia Missiroli, Caterina Boncompagni, Simone Patergnani, Mariasole Perrone, Carlotta Giorgi, Paolo Pinton, Marcello Govoni, Carlo Alberto Scirè, Alessandra Bortoluzzi, Ettore Silvagni

**Affiliations:** 1Rheumatology Unit, Department of Medical Sciences, Università degli Studi di Ferrara, Azienda Ospedaliero-Universitaria S. Anna, 44124 Cona, Italy; 2Department of Precision Medicine, University della Campania L. Vanvitelli, 80138 Naples, Italy; 3Section of Dermatology, Department of Medical Sciences, University of Ferrara, 44121 Ferrara, Italy; 4Anatomic Pathology, Department of Translational Medicine, University of Ferrara, 44121 Ferrara, Italy; 5Department of Medical and Biological Sciences, Rheumatology Institute, University Hospital Santa Maria della Misericordia, 33100 Udine, Italy; 6Laboratory for Technologies of Advanced Therapies, Department of Medical Sciences, Section of Experimental Medicine, University of Ferrara, 44121 Ferrara, Italy; 7School of Medicine, University of Milano Bicocca, 20126 Milan, Italy; 8Epidemiology Unit, Italian Society for Rheumatology, 20121 Milan, Italy

**Keywords:** psoriatic arthritis, targeted therapies, synovial biopsy, fibroblast-like synoviocytes, biological DMARDs, targeted synthetic DMARDs

## Abstract

The aims of this systematic literature review (SLR) were to identify the effects of approved biological and targeted synthetic disease modifying antirheumatic drugs (b/tsDMARDs) on synovial membrane of psoriatic arthritis (PsA) patients, and to determine the existence of histological/molecular biomarkers of response to therapy. A search was conducted on MEDLINE, Embase, Scopus, and Cochrane Library (PROSPERO:CRD42022304986) to retrieve data on longitudinal change of biomarkers in paired synovial biopsies and in vitro studies. A meta-analysis was conducted by adopting the standardized mean difference (SMD) as a measure of the effect. Twenty-two studies were included (19 longitudinal, 3 in vitro). In longitudinal studies, TNF inhibitors were the most used drugs, while, for in vitro studies, JAK inhibitors or adalimumab/secukinumab were assessed. The main technique used was immunohistochemistry (longitudinal studies). The meta-analysis showed a significant reduction in both CD3+ lymphocytes (SMD −0.85 [95% CI −1.23; −0.47]) and CD68+ macrophages (sublining, sl) (SMD −0.74 [−1.16; −0.32]) in synovial biopsies from patients treated for 4–12 weeks with bDMARDs. Reduction in CD3+ mostly correlated with clinical response. Despite heterogeneity among the biomarkers evaluated, the reduction in CD3+/CD68+sl cells during the first 3 months of treatment with TNF inhibitors represents the most consistent variation reported in the literature.

## 1. Introduction

Psoriatic disease is a chronic inflammatory disease characterized by complex clinical heterogeneity, affecting different sites, predominantly skin and nails, as well as peripheral joints, axial skeleton and entheses [1]. A severe dysregulation of several pro-inflammatory immune pathways plays a major role in the pathogenesis of the disease, along with genetic background and environmental factors [2,3]. Immune cells infiltrating target organs cause significant production of pro-inflammatory cytokines, including tumor necrosis factor-α (TNF-α) and interleukin (IL)-1β, IL-6, IL-22, IL-23, IL-17A, and IL-18, which cause perpetuation of inflammation [4]. Chronic synovitis is a key feature of psoriatic arthritis (PsA), and it is characterized by hyperplasia of fibroblast-like synoviocytes (FLS) in the intimal lining layer, inflammatory infiltrates of the synovial sublining, and neo-angiogenesis phenomena. Analysis of PsA synovial membrane has shown the presence of macrophages in the lining layer, while in sublining different types of immune cells can be found (macrophages, mast cells, polymorphonuclear cells, T cells, B cells and plasma cells), producing a wide range of cytokines [2,5]. The natural history of the process described above generates potentially progressive cartilage and bone damage.

In recent years, growing knowledge of pathogenesis has allowed advances in the understanding of molecular pathways underlying psoriatic disease, and new therapies have been licensed [6,7]. Despite the suggestion that the treatment decision in PsA should be adapted to individual patient features, the management of psoriatic disease is far from precision medicine [8,9]. Indeed, there are no validated biomarkers able to predict the response to specific therapies [10], and the choice of the drug depends mostly on disease severity, extra-articular manifestations and endotypes of the disease, prognostic factors, prior treatment history, comorbidities, access to therapy, and patient’s preferences, similarly to other cognate diseases such as rheumatoid arthritis (RA) [11,12,13].

Research is, therefore, moving towards the development of new ways of personalizing treatments, to which the histological analysis of synovial membrane samples may contribute [14,15]. The longitudinal histological analysis of the effects on synovium biopsy samples of disease modifying antirheumatic drugs (DMARDs) and the studies of in vitro exposure of cells or tissues to drugs could advance the most influential effects of DMARDs on psoriatic synovial cells and could be applied to the goal of precision medicine. Supported by technological advances in synovial tissue processing, an increasing number of studies are focusing on cellular and molecular changes of synovium after treatments, with potentially relevant clinical implications [16]. A work gathering and summarizing systematically the latest evidence on this topic has not been published. Therefore, the aims of this systematic literature review (SLR) were (i) to identify the synovial effects of approved biological (b)/targeted synthetic (ts) DMARDs in PsA through the analysis of synovial biopsies, and (ii) to determine if these effects could be considered histological/molecular biomarkers of response to therapy.

## 2. Materials and Methods

This SLR was performed to retrieve biomarkers modifications on synovial membrane and skin biopsies after bDMARD or tsDMARD administration (only drugs approved by the EMA for the systemic management of psoriasis and PsA were evaluated; design of studies evaluated: paired biopsies in longitudinal studies, in vitro studies). Here, we provide methods and data regarding the synovial effect of b/tsDMARDs. The SLR was conducted following the PRISMA 2020 Checklist (Appendix A) [17], and the protocol was registered in PROSPERO (CRD42022304986).

First, the research question was translated into patients, intervention, comparator, outcome, study type (PICOs). The population was defined as patients with psoriasis or active peripheral PsA undergoing synovial biopsy. Intervention was defined as synovial biopsy in studies with bDMARDs or tsDMARDs approved for the systemic management of these conditions; for the outcomes, we evaluated (i) biomarker modifications on synovial membrane biopsies/in vitro cell cultures, and (ii) clinical response to systemic treatment (Appendix A). We included all published studies limited to humans, published in English or Italian, by 16 January 2022. Studies not considering synovial membrane biopsy (e.g., in vitro studies in cell lines or cells purchased from companies) or studies involving drugs different from b/tsDMARDs approved for the management of psoriasis or PsA were not included (Appendix A). A comprehensive search was conducted to find eligible articles in different electronic databases: PubMed, Embase, Scopus, and Cochrane Library (Appendix A).

Records were imported into a bibliographic management software (Zotero) and articles appearing in more than one database were considered only once. Articles were selected based on (i) title and abstract and (ii) full text, by two pairs of independent reviewers (MSC, NS, VV, RB), classifying those studies meeting the inclusion criteria. Any disagreement between the reviewers regarding the eligibility of a particular study was resolved through discussion between reviewers; in case of persistent disagreement, a third referee was consulted (ES). Finally, we scrutinized the reference lists of the identified articles to find additional relevant studies. Data from included articles were extracted in pre-specified forms using a secure electronic data capture database for PsA patients (https://www.redcap.ospfe.it accessed on 16 August 2022) [18], hosted at the University Hospital of Ferrara, including general information on the article (title of publication, study design, year of publication), features of the population (baseline demographics, sample size, country, type of intervention, outcome measures) and, when available, standardized mean difference (SMD). The main outcome of interest was the modification of biomarkers following systemic or in vitro drugs, while, for biomarkers of response [19], the clinical response was considered as outcome. Results were presented in summary of evidence tables. Risk of Bias (RoB) of included studies was assessed using the Newcastle–Ottawa scale (NOS) for cohort and case–control studies [20], and the Cochrane RoB (RoB2) for randomized-controlled trials (RCTs) [21]. Discrepancies between the two pairs of reviewers were discussed with a third reviewer (ES).

Descriptive results of the SLR were reported as mean and standard deviation (SD) for quantitative variables. Categorical variables were described as counts and percentages. Metanalyses were conducted to test variations of CD3+ lymphocytes and sublining CD68+ macrophages following bDMARD treatment in longitudinal studies. Mean and standard deviations were extrapolated from full-text articles. Missing data were imputed from median and range [22]. Raw data were requested from authors for a relevant article, but they were unavailable [23]. Standardized mean difference (SMD) was used as measure of effect to combine total synovial cell count with IHC semiquantitative scores. Cohen’s d was employed to calculate estimated effect sizes and was interpreted using the following conventions: small effect if ≥0.20, moderate effect if ≥0.50, and large effect if ≥0.80. Pooled SMD was estimated using a random-effects model with inverse variance method, and results were graphically presented with forest plots. Cochran’s Q and I^2^, respectively, were used to test and quantify between-study heterogeneity [24]. Publication bias was investigated with qualitative inspection of funnel plots and Begg’s test for assessing asymmetry [25]. Influence analyses were repeated after excluding each study once, while sensitivity analyses were performed by removing papers in which part of the sample also included patients with ankylosing spondylitis or other forms of SpA. Finally, a sensitivity analysis was conducted using unstandardized mean difference (MD) from articles in which a semiquantitative IHC score was adopted. Analyses were performed with Stata14 software (STATA Corporation, College Station, TX, USA) and RStudio© using the package ‘meta’ [26].

## 3. Results and Discussion

### 3.1. Descriptive Results

Out of 3111 non-duplicate articles evaluated, 22 were included (Figure 1), of which 19 were longitudinal and three in vitro, for a total of 365 patients with a mean age (standard deviation, SD) of 45.8 (4.3) years (Table 1). Five studies were RCTs, while the others were observational studies (cohort or case-series). Arthroscopy was the biopsy sampling technique used in almost all works. In longitudinal studies, infliximab, etanercept and adalimumab were the most frequently used drugs, while in in vitro studies the effects of JAK inhibitors (tofacitinib, upadacitinib) on synovial explants and FLS were evaluated, as well as adalimumab and secukinumab in FLS and CD4+ T cell co-cultures. Several laboratory outcomes were assessed. The main technique used in longitudinal studies was immunohistochemistry (IHC), often coupled with histology analysis. The average time for outcome assessment was 11.0 (6.4) weeks (longitudinal studies). Table 2 and Table 3 summarize the main findings of the SLR (longitudinal and in vitro studies), while Table 4 and Table 5 highlight RoB assessment for observational studies and RCTs, respectively.

### 3.2. Longitudinal Studies of TNF Inhibitors (Immunohistochemistry—IHC)

The effect of TNF-alfa inhibitors (TNFis) on synovial tissue using IHC was explored in 13 longitudinal studies, showing a modification in various types of inflammatory cells and molecules. The time of paired biopsy assessment varied between 48 h and 12 weeks. The most important finding was a decrease in CD3+ lymphocytes, observed in 10 studies, despite not always reaching statistical significance [23,29,30,31,32,33,34,35,36,39]. T lymphocytes (CD3+) represent one of the most important cellular populations in PsA synovitis, whose role is highlighted by the enrichment in T-cell-derived cytokines in the synovial fluid and inflamed synovium. Among T lymphocytes, CD4+ and CD8+ markers were globally reduced after treatment, as well as endothelial CD31+ cells and CD68+ macrophages, despite not being consistently reported across studies included in this SLR [23,27,29,31,32,33,34,35,36]. TNFis also interfered with the action of dendritic cells, responsible for antigen presentation and cytokine secretion, causing a reduction in C-type lectin domain family 9 member A (CLEC9A) [39], which is involved in the cross-presentation of antigens to CD8+ T cells [48].

Apart from cellular populations, IHC was adopted to unravel the effect of TNFis on cytokine expression, showing no effects of adalimumab on IL-17A, IL-17F, IL-17RA and IL-17RC [28]. Contrariwise, TNFis significantly reduced matrix metalloproteinases (MMPs) in synovial tissue, as demonstrated with adalimumab on MMP13 levels [23], or with etanercept on MMP3 and MMP9 [34], whereas MMP3 expression significantly differed between responders and non-responders to infliximab and etanercept [35]. Regarding adhesion molecules, a significant decrease in the levels of intercellular adhesion molecule 1 (ICAM-1) on synovial capillaries was observed in patients after combination therapy of infliximab and methotrexate [32], as well as a reduced expression of E-selectin [33] and vascular cell adhesion protein 1 (VCAM-1) [33]. Finally, TNFis demonstrated anti-angiogenic properties through significant reduction of αvβ3-positive neo-vessels [27,32] coupled with a decrease in other markers of neo-angiogenesis such as SDF-1+ vessels, vascular-endothelial growth factor (VEGF) and its receptor KDR/flk-1 (VEGFR-2). On the contrary, angiopoietin-2 showed a significant increase.

The RANK/RANKL/osteoprotegerin (OPG) system, known to play a central role in bone resorption by promoting the maturation and activation of osteoclasts, was not significantly affected after 12 weeks of treatment with TNF-α blockade [30]. In fact, no significant differences were detected in synovial RANKL, OPG, and RANK expression, although a general decrease in the degree of cellular infiltration was observed. Despite the absence of effect on the global study population, TNFα blockade decreased the RANKL expression by FLS in a subset of patients with the best clinical response.

### 3.3. Longitudinal Studies of TNF Inhibitors (Other Techniques)

Other key metabolic or pro-inflammatory processes involved in psoriatic synovitis were assessed by the retrieved articles exploiting laboratory techniques other than IHC. Cellular apoptosis was investigated using terminal deoxynucleotidyl transferase dUTP nick end labeling (TUNEL) and caspase-3 staining, showing no increase in apoptosis following short-term TNFis treatment in longitudinal studies (48 h) [31]. Nuclear factor kappa B (NFκB) proteins, a group of transcription factors involved in inflammatory and immune responses able to interact with other transcription factors, including mitogen activated protein kinases (MAPKs) [49], were evaluated by Lories et al. [37], who analyzed protein expression levels of NFκB and the three main MAPKs, namely extracellular regulating kinase (ERK), the c-Jun-N-terminal kinase (JNK), and p38, before and after etanercept (6 months). By immunofluorescence staining and digital image analysis, etanercept acted by reducing NFκB, ERK, and JNK, but not p38 levels.

A Spanish study conducted by the group of Cañete et al. [27] focused on the process of neo-vascularization in the synovium of psoriatic arthritis patients treated with infliximab after methotrexate (MTX) treatment failure. After 8 weeks, the vascular score showed a significant reduction, paired with IHC modifications suggesting a reduction in neo-vessels, as demonstrated by αvβ3+ markers. Similarly, VEGF and VEGFR-2 mRNA expression decreased, and the expression of Ang-2 increased after treatment. The gross histological evaluation permitted underlining a significant effect of infliximab or etanercept in reducing synovial lining layer thickness [33,34] and the number of blood vessels, along with a downregulation of follicular structures organization [33,38].

Another study, by Collins et al. [40], based on proteomics analysis, aimed at identifying protein expression differences in patients treated with TNFis (either etanercept or adalimumab) that could serve as potential biomarkers of treatment response. In their study, 119 different protein spots changed significantly following 4 weeks with etanercept (including haptoglobin, annexin A2, serum amyloid P, peroxiredoxin 6, serum albumin, Ig kappa chain C, fibrinogen beta chain), and 91 protein spots changed significantly following adalimumab treatment (including haptoglobin, serum albumin, ubiquitin conjugating enzyme E2, annexin A1, A2 and A6, serum amyloid P, heat shock cognate 71 kDa protein, fibrinogen beta chain, pyruvate kinase isozymesM1/M2, collagen alpha 3 and cathepsin B). Moreover, researchers identified 25 proteins that were differentially expressed between “good responders” and “poor responders” to adalimumab (annexin A1 and A2, serum albumin, haptoglobin, apolipoprotein A1, collagen alpha 3, actin, rho-GDP-dissociation inhibitor 2, alpha-1B-glycoprotein, 78 kD glucose-related protein, replication protein A, pyruvate kinase M1/M2, heat shock protein 70 kDa and 71 kDa, vimentin and lamin-B2).

### 3.4. Longitudinal Studies of Non-TNF Inhibitors

Four studies examined the adoption of non-TNF inhibitors in longitudinal studies. In the study by Szentpetery et al. [41], abatacept reduced FOXP3+ Treg expression (IHC) in the synovium over 6 months, an effect not confirmed in skin biopsies. No significant variations were reported for CD3+, CD8 or CD31 expression during the study period, despite a trend towards reduction in CD4+ cells. The study by Fiechter et al. [42] analyzed, in 24 patients with PsA and active knee or ankle arthritis, the IHC changes on synovial biopsy after the use of ustekinumab. After 12 weeks of systemic ustekinumab treatment, there was a numerical decrease in all infiltrating immune cells (CD3+, CD15, CD20) and a significant decrease in sublining CD68+ macrophages, as demonstrated by IHC analysis, as well as a reduction in MMP3 mRNA levels. Despite decreasing PsA synovial inflammation, patients with both clinical and ultrasound remission displayed a persistent synovial cellular infiltrate, suggesting residual histological inflammation under ustekinumab. Synovial TNF expression was unaffected by ustekinumab, as were levels of IL-6, IL-8 and IL-17. Instead, ustekinumab interfered with several chemotaxis and neo-angiogenesis pathways, and it remodulated wnt signaling (pro-chondrogenesis effect) and the PI3K-Akt-mTOR and MAPK-ERK pathways. Van Mens et al. [43], instead, evaluated the impact of secukinumab on the synovial immunopathology of 20 patients with peripheral spondyloarthritis (SpA), of which 13 had PsA. Along with the clinical benefit, at week 12, there was a significant reduction in CD15+ neutrophils and CD68 + sl macrophages as measured by IHC. When qPCR analysis was performed, the authors highlighted a significant reduction in IL-6, MMP3, and CCL20 mRNA expression, but not in IL-8. There was also a significant reduction in IL-17A mRNA, while the expression of TNF was unaffected. Chen et al. [44] also evaluated, through IHC analysis, the content of IL-17A inside the mast cells located in the synovial membrane during secukinumab treatment. While the percentage of all IL-17A-positive cells (non-mast cells) decreased, the IL17A content in mast cells increased.

### 3.5. Meta-Analysis of bDMARD Effect in Longitudinal Studies

Five papers evaluated changes in CD3+ lymphocytes from baseline to follow-up in synovial tissue of patients treated with bDMARDs and included in quantitative analysis [29,31,33,34,41]. Fifty-three patients were included, with a mean (SD) age of 46.5 (5.4) years and a mean (SD) disease duration of 92.4 (40.2) months. Patients were exposed to TNFis in four of the five studies, while in one study they were treated with abatacept [41]. Random-effect meta-analysis revealed a significant reduction in CD3+ count after 4 to 12 weeks of bDMARD treatment (pooled standardized mean difference [SMD]= −0.85, 95% CI [−1.23; −0.47], *p* < 0.0001) (Figure 2). No significant between-study heterogeneity was found (I^2^ = 0%, Q = 1.86, *p* = 0.7614). In the sensitivity analysis, which excluded papers considering also the broader SpA population, the pooled effect size had similar effect direction and magnitude (pooled SMD −0.78, 95% CI [−1.32; −0.23] *p* = 0.0054) [33,34]. Influence analyses were performed after excluding each study once, as well as sensitivity analyses using unstandardized mean difference (MD) from articles in which a semiquantitative IHC score was adopted, with no significant variations from primary analyses. Additional details of sensitivity analyses and publication bias assessment, investigated with qualitative inspection of funnel plots and Begg’s test for assessing asymmetry, are shown in Appendix A.

A random-effect meta-analysis was conducted in five studies in which CD68+ macrophages in the sublining were quantified before and after 4 to 12 weeks of TNFis treatment (no other bDMARD was tested) [27,29,31,33,34]. Mean (SD) age was 47.5 (5.6) years, and mean disease duration was 90.4 (44.4) months. The results were consistent with a significant reduction in CD68+ levels (pooled SMD −0.74, 95% CI [−1.16; −0.32], *p* = 0.0005) without substantial heterogeneity (I^2^ = 22%, Q = 5.13, *p* = 0.2744) (Figure 3). The sensitivity analysis, excluding papers permitting the enrollment of other SpA populations, did not reconfirm a significant decline in CD68 + sl macrophage counts (SMD −0.70, 95% CI [−1.50; 0.09], *p* = 0.0841), with moderate level of heterogeneity (I^2^ = 55.1%, Q = 4.46, *p* = 0.1077) [33,34]. Additional details of sensitivity analyses and publication bias assessment are shown in Appendix A.

### 3.6. In Vitro Studies

Only three studies have assessed the effect of b/tsDMARDs through an in vitro design. The consequences of PsA FLS exposure to Janus kinase inhibitors (JAKis) have been analyzed in two cases (tofacitinib, upadacitinib) [45,46]. In the study by Gao et al. [45], the cells were not exogenously stimulated, and synovial explants were analyzed along with FLS, as opposed to the study of O’Brien et al. [46], which focused on FLS stimulated with oncostatin M (OSM), a known inducer of the JAK/signal transducer and activator of transcription (STAT) pathway. Both studies reported similar effects of JAKis in reducing the migration capacity of FLS. Moreover, using enzyme-linked immunosorbent assay (ELISA), the secretion of IL-6 and monocyte chemoattractant protein-1 (MCP-1) were reduced by currently approved JAKis in synovial explants and FLS cultures, respectively. In the study conducted by O’Brien et al. [46], several JAKis (upadacitinib, baricitinib, peficitinib, filgotinib; only the former is actually approved for PsA management) were evaluated. MCP-1 and IL-6 gene expressions were inhibited by baricitinib and upadacitinib. Furthermore, upadacitinib reduced ICAM-1 gene expression. With a cellular bioenergetic function analysis, a change in FLS overall metabolic profile was also observed, consisting of a rise in mitochondrial respiration processes with a concomitant decline in glycolysis. Apart from these, the study of Gao et al. [45] described a reduction in MMPs secretion in synovial explants and a decrease in IL-8 secretion after exposure to tofacitinib, a finding not confirmed with upadacitinib in FLS cultures. In both PsA explants and PsA FLS, tofacitinib was able to reduce NFkBp65 expression, while it reduced invasion and network formations of FLS.

Another in vitro study, by Xu et al. [47], analyzed the effects of adalimumab and secukinumab on co-cultures of CD4+ T cells and FLS after stimulation with anti-CD3 and anti-CD28 for 72 h. The authors used ELISA and RT-PCR for laboratory readouts, and they found that secukinumab significantly reduced the production of IL-17A and IL-6, whereas adalimumab reduced TNF-alfa, MMP3 and MMP13. Both drugs reduced IL-8 and IL-1b secretion and their mRNA expressions.

### 3.7. Biomarkers of Response to Therapy

Of the included studies, 10 evaluated the clinical response to therapy, none with in vitro design. In three of these studies, all of which considered TNFis, the longitudinal reduction in CD3+ cells predicted the clinical response [23,29,35], while in one study with secukinumab, no correlation was found between changes in clinical scores and changes in cellular infiltrate, including CD3+ [43]. Other studies evaluated different parameters, such as the increase in IL-17A-positive mast cells [44], reduced ectopic lymphoid neogenesis [38], decreased RANKL [30], MMP13 [23] or MMP3 [35] expression, decreased infiltration of macrophages or polymorphonuclear cells [35], or the ratio of differentially expressed genes [42] or proteins [40] in responder patients compared to non-responders, suggesting that these biomarkers could be promising in antedating the effectiveness of systemic DMARDs. Regarding synovial histology, neither lining layer hyperplasia nor vascularity modifications correlated with disease activity over time [35]. Furthermore, no correlation was found between pre- and post-treatment measurements of NFκB and MAPK activation in synovial tissue and disease activity parameters [37].

### 3.8. Discussion 

This SLR aimed to deepen understanding of the synovial mechanisms of action of b/tsDMARDs approved for the systemic management of PsA, trying to evaluate their synovial effects in a standardized manner, exploring how this effect correlates with clinical response. Since no validated biomarker has yet entered clinical practice [10], the full knowledge of the synovial impact of such drugs is of value. This is particularly true since it is believed that identifying the appropriate treatment in the early stages of the disease is effective in achieving clinical remission, thus avoiding disease progression [8,50]. Our SLR highlighted a significant heterogeneity in laboratory techniques adopted to investigate the effects of DMARDs, with IHC emerging as the most used in longitudinal studies. Longitudinal reduction in CD3+ lymphocytes and CD68+ macrophages of the sublining represents the most common synovial modification following TNFis, with the first parameter emerging as a promising candidate for systemic treatment response prediction. However, the synovial effect of DMARDs in PsA is more wide-ranging than a mere anti-inflammatory effect (Figure 4).

Our SLR aimed to identify two different types of studies, namely longitudinal and in vitro studies. In longitudinal studies (19 studies included), infliximab, etanercept and adalimumab were the most used drugs, while, for in vitro studies (three articles retrieved), tofacitinib and upadacitinib were investigated in FLS cultures, as well as in synovial explant cultures, and adalimumab and secukinumab in co-cultures of FLS and CD4+ T cells. Given this discrepancy, most of the evidence arises for TNFis and from longitudinal studies. Patients enrolled in the included studies were either naïve or previously exposed to bDMARDs, had a moderately long disease duration, and had mainly undergone arthroscopic synovial biopsy, with no US-guided synovial biopsy procedure performed. Since US-guided synovial biopsy is emerging as the technique of choice for synovial membrane analysis in chronic inflammatory arthritis [51,52,53], given its wide availability, the adoption of this technique is expected to increase in the following years, in line with similar experiences in longitudinal studies in RA [54,55].

Focusing on longitudinal studies, our meta-analysis suggested a significant downmodulating effect of several bDMARDs, mostly TNFis, in reducing CD3+ lymphocytes after 4–12 weeks of systemic treatment. This net effect remained significant considering only works assessing the cellular infiltrate in a semi-quantitative manner, and, similarly, after excluding works enrolling patients suffering from other forms of spondyloarthritis. When the same reasoning was applied to CD68 + sl variation, statistical significance was lost. In our opinion, this distinction applies specifically to PsA patients, and it might contribute to the distinction of PsA from RA, for which it is well-known that the reduction in CD68 + sl is one of the most well-characterized readouts of an effective treatment, resistant to placebo effects [56,57,58,59]. Moreover, longitudinal CD3+ reduction was the most frequently assessed modification related to systemic treatment response, suggesting this modification can be explored as a response biomarker. Regarding the design of the studies, it must be underlined that longitudinal studies carry some relevant limitations; the most relevant relates to the necessity for a patient to undergo separate biopsy procedures. This could be challenging for patients experiencing significant symptom relief following therapy, with a low rate of potential impact on clinical practice [60,61]. To this end, as demonstrated in patients with RA in clinical remission, the possibility of exploiting a less-invasive US-guided synovial biopsy procedure might be valuable [62]. Timelines for repeated biopsy assessment are not validated, and the numbers of drugs tested in these studies were necessarily low. Formally, a variation in a response biomarker should antedate the response to treatment; therefore, the utility of assessing synovial response after 12 weeks could be questioned. Moreover, synovial histological patterns, well-defined for RA [59,63], were not explored in PsA studies, and semi-quantitative scoring systems, advocated by OMERACT [64] and EULAR [65] to be used in synovial tissue research, were not homogeneously described in the included studies, with a significant heterogeneity in reporting that contributed to the low generalizability of the results, with no application of Krenn’s synovitis score [66].

As previously said, apart from the reduction in cellular components of the synovial inflammatory burden using IHC, longitudinal studies explored other relevant effects of bDMARDs, combining IHC with other relevant laboratory techniques, elucidating a more complex mechanism of action. Histologic evaluation was used to study the process of neovascularization in the synovium, affected by systemic drugs, as well as their effects on synovial lining layer thickness. As suggested by some authors [34], an effective treatment is mostly involved in modulation of ongoing inflammation in the short term, while the structural recovery might occur after a more prolonged treatment duration, and this cannot always be the case [67]. While apoptosis of cellular elements seemed not to be affected by TNFis in short-term studies, several reports pointed towards a reduction in MMP release and synthesis, expression of adhesion molecules such as ICAM-1, VCAM-1, E-selectin, responsible for white blood cell diapedesis and chemotaxis, NFκB and MAPK signaling, and pro-inflammatory cytokines transcription using bDMARDs, not restricted to TNFis. These aspects underline an environment with reduced cell trafficking, cell migration and/or tissue infiltration, possibly leading to less structural damage. It must be noted that bulk mRNA analysis was applied in the totality of studies assessing mRNA expression, with no data regarding single cell RNA sequencing [68]. The effect of DMARDs at the single cell level should be investigated in future studies.

With in vitro studies, a net effect towards a reduction in cytokines or chemokines release mediated by JAKis or bDMARDs in FLS or synovial explant cultures was enhanced [45,46,47], with anti-invasive and anti-migratory actions on FLS, coupled with a metabolic shift from glycolysis to a more aerobic oxidative phosphorylation [45,46], similarly to that demonstrated in RA [69]. As a matter of fact, in vitro studies, especially when they involve non-homogeneous adoption of drug concentrations and time exposures, remain mostly mechanistic, and the real impact on generalizability of the results should always be fully considered.

Given these observations and reasonings, the degree to which these kinds of works can have clinical impact in the mid-term remains to be fully elucidated. As underlined, both types of studies assessed (longitudinal and in vitro) face intrinsic limitations that prevent their full clinical application and, ideally, a biomarker of response should be retrieved before starting a systemic treatment, not after its adoption. To this end, current predictive approaches based on baseline synovial features/biomarkers (not addressed in this SLR) have failed in identifying biomarkers of response applicable to clinical practice in chronic inflammatory arthritis [10], but new and important results are emerging in RA [70,71], and it is expected that positive flags will follow also in the context of PsA. Moreover, only few articles reported data on paired synovial and skin biopsies, showing contradictory results regarding a possible unidirectional trend in the reduction of CD3+ cells in different tissues [32,41]. Whether the effect of b/tsDMARDs is directed primarily towards the synovium instead of other immune-competent organs is still under investigation. 

Our work has some limitations. For some articles, the cohort of patients was not restricted to PsA patients, since enrollment was allowed for subjects also suffering from other types of spondyloarthritis different from PsA, and separation of the results according to disease sub-categories was not always possible. However, we performed sensitivity analysis in our meta-analysis excluding studies involving SpA patients, underlining an effect that was even more specific to PsA for CD3+ cells. Full articles published after the deadline of our SLR did not vary substantially from the conclusions inferred by our SLR [72]. We also excluded animal models from our review, as well as cell lines, FLS purchased from companies, or synovial fluid analyses, and we did not focus on studies assessing remission under bDMARD treatment in which no baseline biopsy was performed [67]. Moreover, we did not focus on entheseal biopsies under b/tsDMARD treatment, and we assessed in longitudinal studies only drugs administered using the routes of administration formally approved by the EMA (e.g., intra-articular administration of TNFis was excluded) [73,74]. However, our work has some relevant strengths, as, to the best of our knowledge, this is the first SLR assessing the synovial effect of approved b/tsDMARDs in PsA, and the addition of a meta-analysis quantifies and strengthens the most important variations observed.

## 4. Conclusions

Despite a high heterogeneity among the biomarkers evaluated, the reduction in CD3+ and CD68 + sl during the first 3 months of treatment with bDMARDs, mostly, but not restricted to, TNFis, represents the most consistent variation reported in the literature, but the synovial effect of DMARDs is much more variable than a pure anti-inflammatory effect. The possibility of easily collecting synovial material using US-guided synovial procedures might increase the information regarding the effect of the always-increasing number of DMARDs approved for the systemic management of this condition, and the possibility of comparing several drugs in head-to-head studies, even considering other key extra-synovial involvements such as skin and entheses, exploiting international collaborations and more sophisticated laboratory techniques or data synthesis methods, can change the actual knowledge of the synovial effects of these drugs and the actual treatment paradigm of a "trial and error" approach in psoriatic disease management.

## Figures and Tables

**Figure 1 ijms-24-05006-f001:**
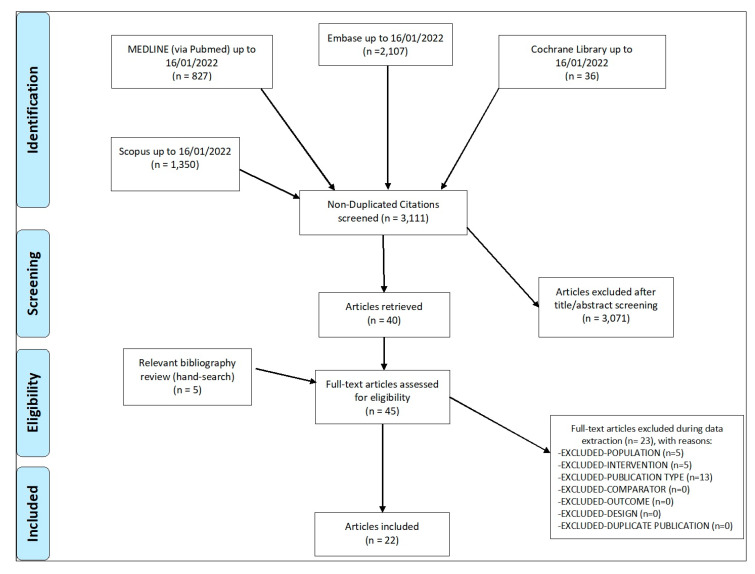
PRISMA Flowchart. Abbreviations: PRISMA: Preferred Reporting Items for Systematic Reviews and Meta-Analyses.

**Figure 2 ijms-24-05006-f002:**
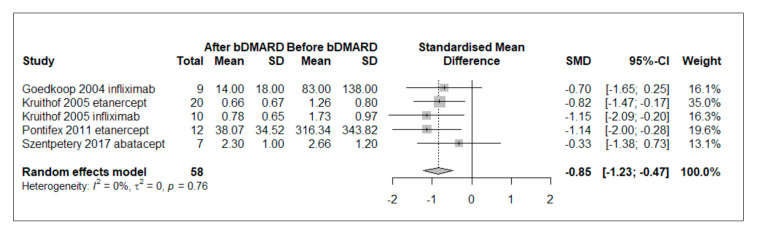
Forest plot of standardized mean difference of CD3+ lymphocytes in synovial tissue of PsA patients treated with bDMARDs. Negative effect means reduction in CD3+ lymphocyte level. Abbreviations: bDMARD: biological disease-modifying anti-rheumatic drug, SD: standard deviation, SMD: standardized mean difference; 95% CI: 95% confidence interval.

**Figure 3 ijms-24-05006-f003:**
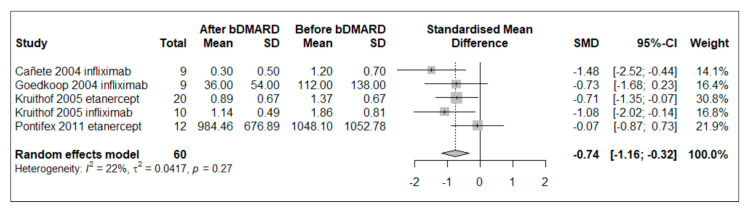
Forest plot of standardized mean difference (SMD) of CD68+ macrophages in sublining synovial tissue of PsA patients treated with bDMARDs (TNFis). Negative effect means reduction in CD68+ macrophage level. Abbreviations: bDMARD: biological disease-modifying anti-rheumatic drug, SD: standard deviation, SMD: standardized mean difference; 95% CI: 95% confidence interval.

**Figure 4 ijms-24-05006-f004:**
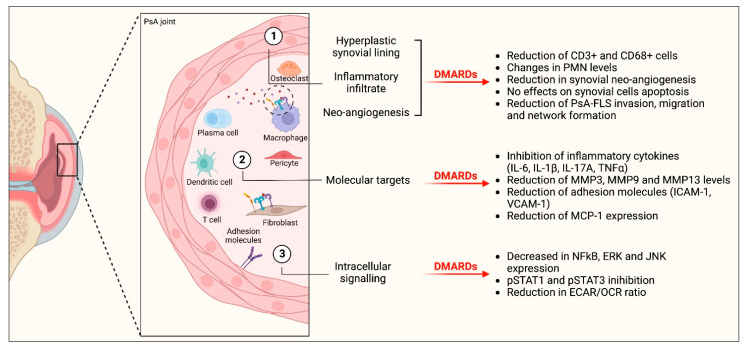
Pictorial view of the main synovial effects of approved b/tsDMARDs retrieved in the literature. Abbreviations: PMN: polymorphonuclear cells, PsA: psoriatic arthritis, FLS: fibroblast-like synoviocytes, IL: interleukin; TNFα: tumor necrosis factor alfa; ICAM-1: intercellular adhesion molecule 1, VCAM-1: vascular cell adhesion protein 1, MMP: matrix metalloproteinases, MCP-1: monocyte chemoattractant protein-1, NFKB: nuclear factor kappaB, ERK: extracellular signal-regulated kinases, c-Jun N-terminal kinases, STAT: signal transducer and activator of transcription, ECAR/OCR: extracellular acidification rate/oxygen consumption rate ratio. This image was created with ©BioRender 2021.

**Table 1 ijms-24-05006-t001:** Main characteristics of the 22 included studies.

Variable	Frequency
Study type	
In vitro studies, N (%)	3 (13.6)
Longitudinal studies, N (%)	19 (86.4)
Characteristics of population and disease	
Number of participants, mean (SD)	17.2 (10.5)
Number of female participants, mean (SD)	8.3 (5.5)
Age, mean (SD)	45.8 (4.3)
PsA disease duration in months, mean (SD)	88.0 (40.8)
PsO disease duration in months, mean (SD)	234 (25.5)
Duration of follow-up in weeks, mean (SD)	11.0 (6.4)
Drug tested	
Infliximab, N (%)	8 (36.4)
Etanercept, N (%)	7 (31.8)
Adalimumab, N (%)	5 (22.7)
Secukinumab, N (%)	3 (13.6)
Ustekinumab, N (%)	1 (4.6)
Tofacitinib, N (%)	1 (4.6)
Upadacitinib, N (%)	1 (4.6)
Abatacept, N (%)	1 (4.6)
Synovial biopsy	
Site of biopsy: Knee, N (%)	17 (77.3)
Site of biopsy: Wrist, N (%)	5 (22.7)
Site of biopsy: Ankle, N (%)	5 (22.7)
Technique: Arthroscopic synovial biopsy, N (%)	20 (90.9)
Technique: Blind needle biopsy, N (%)	1 (4.6)
Technique: US-guided, N (%)	0
Technique: Arthrotomy, N (%)	0
Type of laboratory technique assessed	
Immunohistochemistry, N (%)	17 (77.3)
mRNA extraction and RT-PCR, N (%)	6 (27.3)
ELISA, N (%)	5 (22.7)
Optical microscopy analysis, N (%)	7 (31.8)
Immunofluorescence staining, N (%)	3 (13.6)
Western blot, N (%)	2 (9.1)
TUNEL assay, N (%)	1 (4.6)
Migration, invasion, Matrigel network, N (%)	2 (9.1)
Cellular bioenergetic function analysis, N (%)	1 (4.6)
Gene ontology analysis, N (%)	1 (4.6)
Mass spectrometry, N (%)	1 (4.6)
Cellular population assessed	
FLS or subpopulations, N (%)	4 (18.2)
T cells or subpopulations, N (%)	13 (59.1)
B cells or subpopulations, N (%)	8 (36.4)
Macrophages or subpopulations, N (%)	11 (50.0)
Mast cells or subpopulations, N (%)	2 (9.1)
Endothelial cells or subpopulations, N (%)	5 (22.7)
Synovial explants, N (%)	1 (4.6)

Abbreviations: SD: standard deviation, PsA: psoriatic arthritis, PsO: psoriasis, mRNA, messenger ribonucleic acid, rt-PCR: reverse transcription polymerase chain reaction, ELISA: enzyme-linked immunosorbent assay, TUNEL: terminal deoxynucleotidyl transferase dUTP nick end labeling, FLS; fibroblast-like synoviocytes.

**Table 2 ijms-24-05006-t002:** Summary of Evidence (longitudinal studies).

Article	Year	Study Design	Number of Participants	PsA Disease Duration (Mean Months)	Drug Adopted	Mean Duration of the Follow-Up (Main Outcome–Weeks)	Type of Laboratory Technique Assessed	Main Results
van Kuijk A [23]	2009	RCT	24	66	adalimumab	12	IHC	↓ CD3+-positive T cell infiltration and expression of MMP13.
Cañete [27]	2004	Cohort	9	120	infliximab	8	IHC Histologic evaluation mRNA extraction/RT-PCR	↓ CD68 + sl macrophages ↓ SDF-1 in vessels, neovessels and VEGF, VEGFR-2 ↓ vascular score in sequential biopsies ↓ VEGF and VEGFR2 mRNA expression and increase in Ang-2
Bolt [28]	2021	Cohort	12	106	adalimumab	4	IHC	no change in IL- 17A, IL-17F, IL-17RA and IL-17RC
Pontifex [29]	2011	Cohort	15	90	etanercept	12	IHC	↓ CD3+ T cells in responders to etanercept
Vandooren [30]	2008	Cohort	11	60 (median)	etanercept, adalimumab	12	IHC	no changes in synovial RANKL, OPG, and RANK expression
Goedkoop [31]	2004	RCT	12	-	infliximab	0.2	IHC TUNEL assay	↓ total number of T cells and CD68 no effect of TNFis on synovial cell apoptosis
Goedkoop [32]	2004	Cohort	11	108	infliximab	4	IHC	↓ CD3+ T cells and CD68+ macrophages in synovial tissue ↓ synovial neo-angiogenesis (downregulation of both vWF and αvβ3-positive vessels, VEGF) ↓ adhesion molecules (ICAM-1)
Kruithof [33]	2005	RCT	13 for study population I; 20 for study pop. II	168 for study population I; 222 for study population II	infliximab	12	IHC Histologic evaluation	↓ neutrophils, macrophages, and T cells, but not B cells ↓ synovial lining layer thickness, endothelial activation and number of blood vessels, downregulation of follicular structures organization
Kruithof [34]	2005	Cohort	20	60	etanercept	52	IHC Histologic evaluation	↓ global cellular infiltration, T lymphocytes, macrophages subsets (CD68+, CD163, MRP-8, and MRP-14) ↓ lining layer hyperplasia, moderate reduction in vascularity ↓ synovial expression of MMP3 and MMP9
Kruithof [35]	2006	Cohort	20 treated with infliximab 20 treated with etanercept 12 controls	52	infliximab and etanercept	12	IHC Histologic evaluation	changes in synovial macrophage subsets, PMN levels, and MMP3 expression in responders to treatment vs. non-responders no significant difference between responders and non-responders regarding synovial histology modification
de Rycke [36]	2005	Cohort	40 with SpA	78	infliximab	12	IHC Histologic evaluation	↓ infiltrating macrophages as demonstrated by the change in the expression of myeloid-related protein in both the lining layer and the sublining layer (MRP-4, MRP-8)
Lories [37]	2008	Case-series	9	109.3	etanercept	26	Histologic evaluation Immunofluorescence staining	↓ lining layer hyperplasia and normalization of sublining vascularity. ↓ cell infiltration and lymphoid follicles ↓ NFkB, ERK, JNK but no variation in p38
Canete [38]	2007	Cohort	27	82.6	infliximab	12	Histologic evaluation IHC	↓ ectopic lymphoid neogenesis
Ramos [39]	2016	RCT	24	-	adalimumab	4	ICH, mRNA extraction/RT-PCR	↓ CLEC9A expression in PsA synovial tissue in the adalimumab treated group compared with placebo after 4 weeks
Collins [40]	2016	Cohort	12	-	etanercept, adalimumab	12	Mass spectrometry	119 different protein spots changed significantly (etanercept treatment) including haptoglobin, annexin A2, serum amyloid P, peroxiredoxin 6, serum albumin, Ig kappa chain C, fibrinogen beta chain 91 different protein spots changed significantly (adalimumab treatment) including haptoglobin, serum albumin, ubiquitin conjugating enzyme E2, annexin A1, A2 and A6, serum amyloid P, heat shock cognate 71 kDa protein, fibrinogen beta chain, pyruvate kinase isozymes M1/M2, collagen alpha 3 and cathepsin B
Szentpetery [41]	2017	RCT	15	120	abatacept	8	IHC Dual IF staining	↓ FOXP3+ T-cells in the synovium = CD4+, CD3+, CD31+, CD8+
Fiechter [42]	2021	Cohort	11	-	ustekinumab	24	IHC mRNA extraction RT-PCR Gene ontology analysis	↓ CD68 + sl at 12 weeks, but not at 24 weeks ↓ expression of IL23A mRNA at 12 weeks but not at 24 weeks ↓ MMP3 levels at 12 weeks interference with chemotaxis and neo-angiogenesis pathways, wnt-signaling and PI3K-Akt-mTOR and MAPK-ERK pathways
van Mens [43]	2018	Cohort	20	66 (median)	secukinumab	12	IHC mRNA extraction/RT-PCR	↓ CD15+ neutrophils and CD68 + sl macrophages ↓ synovial mRNA expression of IL-6, MMP3, CCL20 and IL-17A; = synovial mRNA expression of TNF
Chen [44]	2019	Cohort	15	--	secukinumab	12	IHC	↑ IL-17A-positive mast cells ↓ all IL-17A-positive cells (non-mast cells)

Abbreviations: PsA: psoriatic arthritis, RCT: randomized-controlled trial, IHC: immunohistochemistry, MMP: matrix metalloproteinases, mRNA, messenger ribonucleic acid, rt-PCR: reverse transcription polymerase chain reaction, VEGF: vascular endothelial growth factor, IL: interleukin, RANK-L: receptor activator of nuclear factor κ B ligand, OPG: osteoprotegerin, TUNEL: terminal deoxynucleotidyl transferase dUTP nick end labeling, vWF: von Willebrand factor, ICAM-1: intercellular adhesion molecule 1, FOXP3: forkhead box P3. ↓: reduction, ↑: increase; = no variation.

**Table 3 ijms-24-05006-t003:** Summary of Evidence (in vitro studies).

Article	Year	Study Design	Number of Participants	PsA Disease Duration (Mean Months)	Drug Adopted	Type of Laboratory Technique Assessed	Main Results
Gao [45]	2016	Cohort	11	-	tofacitinib	Western blot ELISA Invasion, migration and Matrigel network formation assays	↓ pSTAT1, pSTAT3 (FLS), NFkBp65 (FLS, synovial explants) ↑ SOCS3 and PIAS3 (FLS) ↓ IL-6, IL-8, MCP-1, MMP3, MMP2/9 (synovial explants) ↓ PsA-FLS invasion, migration, network formation.
O’Brien [46]	2021	Cohort	14	11.7	upadacitinib	ELISA mRNA extraction and RT-PCR Invasion, migration assays Cellular bioenergetic function analysis	↓ expression of IL-6, MCP-1, and ICAM-1 (FLS) ↓ migration capacity of PsA FLS ↓ ECAR/OCR ratio
Xu [47]	2020	Cohort	20	-	secukinumab, adalimumab	ELISA and RT-PCR (co-cultures of CD4+ T cells and FLS)	↓ IL- 6 and IL- 1β after secukinumab/adalimumab ↓ TNF, MMP3 and MMP13 after adalimumab ↓ IL-17A and IL-6 after secukinumab.

Abbreviations: PsA: psoriatic arthritis, STAT: signal transducer and activator of transcription, NFKB-p65: nuclear factor κ B protein 65, FLS: fibroblast-like synoviocytes, IL: interleukin, MMP: matrix metalloproteinases, MCP-1: monocyte chemoattractant protein-1, ICAM-1: intercellular adhesion molecule 1, ECAR/OCR: extracellular acidification rate/oxygen consumption rate ratio, ELISA: enzyme-linked immunosorbent assay, rt-PCR: reverse transcription polymerase chain reaction, ↓: reduction; ↑: increase; = no variation.

**Table 4 ijms-24-05006-t004:** Newcastle–Ottawa scale for risk of bias assessment for case–control and cohort studies [20]. Stars identify high quality choices.

	Selection	Comparability	Exposure/Outcome	Total
Bolt, 2021 [28]	★★	-	-	2
Cañete, 2007 [38]	★★	★	★★	5
Cañete, 2004 [27]	★★★★	★★	★★★	9
Chen, 2019 [44]	★	-	★★	3
Collins, 2016 [40]	★★	-	★★	4
de Rycke, 2005 [36]	★★	★★	★	5
Fiechter, 2021 [42]	★★★★	★	★★★	8
Gao, 2016 [45]	★★★★	★★	★★★	9
Goedkoop, 2004 [32]	★★★	-	★★★	6
Kruithof, 2005 [34]	★	-	★★	3
Kruithof, 2006 [35]	★★★	-	★★	5
Lories, 2008 [37]	★★★	-	★★★	6
O’Brien, 2021 [46]	★★	★	★★	5
Pontifex, 2011 [29]	★★★	-	★★★	6
van Mens, 2018 [43]	★★	-	★★	4
Vandooren, 2008 [30]	★★	-	★	3
Xu, 2020 [47]	★	-	★★	3

**Table 5 ijms-24-05006-t005:** Cochrane RoB 2 for risk of bias assessment for randomized controlled trials [21]. Green circle with cross indicates low risk of bias. Yellow circle with question mark indicates some concerns about risk of bias. Red circle indicates high risk of bias. Abbreviations: RoB: risk of bias.

	RandomizationProcess	Deviations fromIntended Interventions	Missing OutcomeData	Measurement ofthe Outcome	Selection of theReported Result
Goedkoop, 2004 [31]	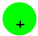	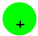	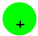	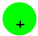	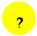
Kruithof, 2005 [33]	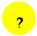	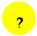	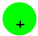	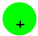	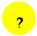
Ramos, 2016 [39]	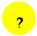	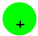	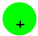	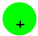	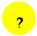
Szentpetery, 2017 [41]	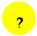	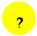	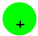	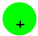	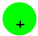
van Kuijk, 2009 [23]	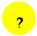	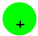	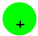	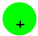	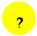

## Data Availability

All data are available from the authors upon specific request.

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
