# Peer review of "Evaluation of the Synovial Effects of Biological and Targeted Synthetic DMARDs in Patients with Psoriatic Arthritis: A Systematic Literature Review and Meta-Analysis"

_ijms, 2023, doi:10.3390/ijms24055006_

Round 1

Reviewer 1 Report

The Systematic review “Evaluation of the synovial effects of biological and targeted (...) " by Ciliento at al is well written, data analysis is carefully done and tables are a good complement to the text.

Reading this manuscript, it is easy to recognize the need to create interactions between groups/structures in order to develop systematic studies able to gather high number of patients, and analyzed them with shared detailed protocols.

I don’t have major points to be addressed before publication.

I invite authors to revise the text to correct few miss tapping mistakes.

Author Response

We are grateful to this Reviewer for this comment. It is true that the creation of large and multicenter collaborations represents one of the major unmet needs in the field, and we really hope our paper can contribute to this end. We carefully read the manuscript again and modified some orthographic mistakes.

Reviewer 2 Report

The review is a multi centred longitudinal  report comprehensively explaining  the effects of approved biological and targeted synthetic disease modifying antirheumatic drugs on synovial membrane of psoriatic arthritis (PsA) patients and available/reported histological/molecular biomarkers of response to therapy.

1. Authors have mostly covered all the data base to comprehensively explain the review. However, pubmed cited reference should be included to give more coverage for the study

2. Data presentation seems to be outstanding

3. Abstract sections needs to be re-structured to match with a review article style

4. In the Discussion, in addition to summarize the available literature, authors have to include their view points/explanation to support or reject the context/approach.

Author Response

We would like to thank this Reviewer for the comments. We modified the “Abstract” section trying to render it more fluent, however our article is not a narrative review, therefore the structure remained, at least partially, schematic. Moreover, we included a brief paragraph in the “Discussion” chapter to give our viewpoint on the topic, also answering to a similar comment by Reviewer 3, and, similarly, we cautiously shortened the “Results” section. However, we are not fully sure to have understood the first point raised by this reviewer. In fact, we performed a systematic literature review analyzing a specific set of inclusion and exclusion criteria (Supplementary Material 1.2) in different electronic databases, included PubMed. Only the bibliographic entries satisfying these criteria were included for full text analysis. These points are already explained in the “Materials and Methods” section. Moreover, all the bibliographic entries of the systematic literature review are cited accordingly in our paper.

Reviewer 3 Report

Please accept the review once the authors address the comments especially on the discussion section.

The review is a systematic literature review/report of the effects of clinically approved drugs (such as Bio-DMARDS and tsDMARDs) for the PsA patients focusing on the synovial membrane and the authors have reported histological and molecular biomarkers of response to therapy.

1. Authors have done a diligent job of compiling the data from different databases and have given an excellent report of the same.

2. Data representation and figures are clear and well represented.

3. Discussion does not include the authors perspective of their finding. For eg; authors could briefly talk about how their finding might help with the treatment algorithm of PsA patients or help with the unmet medical need of PsA. They could also address on how they think the reduction of CD3 and CD68 post TNFi may contribute to the effect the drug has on the other domains in the PsA patients

Author Response

We are grateful to this Reviewer for these comments. We expanded the “Discussion” section adding a brief paragraph with our viewpoint on the main findings of the review and their potential clinical implications, as well as a possible explanation of the relationships among different domains in response to effective treatments.